# The Impact of Severe ME/CFS on Student Learning and K–12 Educational Limitations

**DOI:** 10.3390/healthcare9060627

**Published:** 2021-05-25

**Authors:** Faith R. Newton

**Affiliations:** Department of Education, Delaware State University, Dover, DE 19901, USA; fnewton@desu.edu

**Keywords:** chronic illness, housebound, chronic fatigue syndrome, myalgic encephalomyelitis, severely ill

## Abstract

Children with ME/CFS who are severely ill are bedbound and homebound, and oftentimes also wheelchair-dependent. Very seriously affected children are often too sick for doctor’s office visits, let alone school attendance. The most recent data estimate that 2–5% of children may be severely affected or bedridden. However, there is no recent research that confirms these numbers. The severely ill receive little help from their schools, and are socially isolated. This article outlines several suggestions for the type of education that students with ME/CFS should be receiving and develops a preliminary sketch of the web of resources and emergent techniques necessary to achieve these outcomes.

## 1. Article

Educators and clinicians focused on children suffering from mild-to-moderate ME/CFS have made significant strides in the past decade in supporting chronically ill students in achieving academic success. Classroom teachers and other education professionals are increasingly designing effective Individual Educational Plans (IEPs), creating supportive classroom learning environments, and modifying curricula to emphasize mastery over completion [1]. Many treating physicians effectively collaborate with schools to unify clinical and instructional best practices, becoming effective allies for students and families [2,3]. Work remains to refine successful techniques and spread their implementation, but a foundational template for educating children with ME/CFS has been achieved [4,5].

The same cannot be said regarding the education of children whose ME/CFS symptoms are so severe as to render them housebound and/or bedbound. Despite sometimes heroic measures by family members, caregivers, and visiting teachers, children with ME/CFS often face spending years in social and intellectual isolation, devoid of challenging cognitive stimulation or the opportunity to learn new skills and information. Though estimates remain little more than guesses because bedbound students are rarely disaggregated in reported statistics, it is likely that thousands of children, adolescents, and young adults suffering from ME/CFS are essentially being “warehoused” in their own bedrooms [6,7,8]. Pendergrast et al. [9] stated in a study of adult housebound patients with ME/CFS that ”25% or more are confined to their homes (housebound) or completely bedbound.” Although, there are no studies outlining the percentages of children, it would not surprise this author if the numbers were as high. This condition commands our attention because there is evidence that children with ME/CFS can learn and lead far more enriched lives whether or not they are physically capable of leaving their bedrooms. Innate human dignity requires us to explore, develop, and execute strategies for alleviating this situation.

This article seeks to accomplish three modest goals:(1).delineate the impact of severe pediatric ME/CFS on student learning, and how that impact inherently limits the positive impact that even the most innovative K–12 educational programs can expect to achieve;(2).establish an initial foundation—in philosophical and process terms—for the *type* of education that students with ME/CFS *should be* receiving, and the educational outcomes to be expected; and(3).develop a preliminary sketch of the web of resources and emergent techniques necessary to achieve these outcomes.

## 2. The Impact of Severe ME/CFS on Student Learning and K–12 Educational Limitations

Myalgic Encephalomyelitis/Chronic Fatigue Syndrome (ME/CFS) is characterized by profound, medically unexplained fatigue including a range of sleep, pain, cognitive, neuroendocrine, and immune symptoms. Post-exertional malaise (the provocation of fatigue by physical, cognitive, or orthostatic stress resulting in symptoms including cognitive fogginess, headache, light-headedness, flu-like symptoms, and muscle aches) and persistent generalized fatigue make school attendance irregular even in mild-to-moderate cases; the most common feature of pediatric ME/CFS observable in the classroom is the array of attention, concentration, and memory deficits known collectively as “Brain Fog” [10]. Children with ME/CFS often view school as a frustrating, painful experience, a situation further complicated by symptom variability. Some students will have only mild symptoms requiring modest accommodations, while others will present with more severe and debilitating symptoms. Children with moderate-to-extreme cases will be homebound or bedbound, resulting in protracted or even permanent non-attendance at school [3,11]. Speight pointed out that the muscular, gastric, fatigue-based, neural, and cognitive symptoms in severe ME/CFS “can actually be worse than that suffered by patients with other chronic conditions such as multiple sclerosis and cancer,” and that “abdominal pain may be so severe as to interfere with nutrition, and in some cases are due to an added complication, Mast Cell Activation Disorder” [12].

Mental health consequences are equally profound, as Boularzreg and Roach note, with catastrophism, pessimism, depression, and isolation, leading to “a substantial loss of self” because “adolescents with ME/CFS attach significance to attending school and hang-outs with friends, [and] when deprived of these events, the adolescent questions the meaning of life” [13]. As one bedbound patient noted:

“Imagine having the flu, being severely jet-lagged, and having not slept for two days but without the sinus and lung congestion—that’s the closest I’ve found to a description of what this illness is like, but it understates it. I lack mental clarity; I mix up words, and I have memory problems and trouble focusing. I have an enormous need for sleep, which never refreshes. There is an overwhelming, permanent, and intense malaise” [11].

A mother of two sons in their mid-twenties who have been homebound or bedbound for a decade reports that they often cannot be upright in bed, and that severe light, sound, and touch sensitivities restrict their use of computers. Their attention spans are often limited to irregular intervals 20 min long, and while “their intellect is not impaired, the [physical] resources available to apply that intellect are severely limited.”

American public schools are chronically ill-equipped to educate such children, adolescents, and young adults [14,15]. Educators lack training for dealing with chronically ill children in their classrooms, let alone those permanently isolated at home [16,17]. Homebound instruction is governed not by consistent national standards and best practices, but by state statutes and school district policies. These often have time-consuming bureaucratic requirements to qualify for services (sometimes mandating re-qualification every three months); severely limiting both the availability of tutors (who may have no formal training for teaching chronically ill children); and tie all education to the courses the student would be taking were s/he actually attending school [18,19,20]. Additional barriers are imposed by an unwillingness to extend deadlines; the expectation that services will be duration-limited and focused on eventual “re-entry” into the regular classroom; and no responsibility for providing educational services past age 21 [21]. Leaving aside nationally specific legal requirements, similar situations pertain internationally; there is significant literature pertaining to these issues in Australia, the Czech Republic, India, New Zealand, Japan, Turkey, across the European Union, and elsewhere [4,22,23,24,25,26,27,28].

## 3. Why Children with Severe ME/CFS Must Become Semi-Independent Lifelong Learners

Most innovations introduced to support the education of bedbound individuals suffering from severe ME/CFS have been technology-based. Robotics are intended to allow these students virtual access to live classrooms in a real-time, interactive three-dimensional setting [29]. New kinds of bed tables or trays have been developed to allow more comfortable access to computers. An increasingly broad array of fully online courses has become available, with their use expanding beyond the original purpose of credit recovery to what is often a default choice suggested for homebound students [30].

These innovations are intended as aides to making the existing patchwork system function, rather than rethinking the philosophy, organization, and objectives of educating bedbound students. They are sometimes perceived by school districts as comparatively inexpensive and not requiring significant investments of staff or resources, and often lump all bedbound students into a single category. A robotic telepresence is of extremely limited usefulness to a child with ME/CFS whose working attention span for active learning exists in irregularly spaced 20 min intervals, and who has enough issues with processing speed without having to operate a complex interface while trying to pay attention to a lecture or group activities. Many of the solutions to these issues are being pursued in hospitals, not schools [31].

In fact, for decades, the most innovative work in educating bedbound students (and older patients as well) has originated in hospitals providing long-term care rather than the educational system. In 1942, Dr. Elena D. Gall of Hunter College established a continuing education program at Goldwater Memorial Hospital in New York City, which had a population of 1850 patients permanently confined to a hospital setting. Designed in 1937 by architect Isadore Rosenfield, who “was as concerned with the careful design of a patient’s bedside lamp as he was with the circulation patterns of the thousands of people who would use the facility each day,” Goldwater Memorial embodied an “unwavering patient-centric design approach” [32].

Gall theorized that the patients “are thinking, intelligent individuals who still function on an intellectual level,” but that “agencies which offer continued education to interested adults did not reach out to people who were to be hospital bound for many years.” She created a continuing education program that was multigenerational, democratically run by the patients themselves, and focused on lifelong learning rather than specifically tied to credentialing within the public education system: “The patients, ranging from teenagers to elderly people, meet every Friday night in one of the recreation rooms of the hospital. Each year they themselves choose the area of interest for the term of study. One year it was English literature, another time it was world affairs. This year they decided on current events. They have their own roster of duly elected officers” [33,34]. This program appears to have operated successfully until at least 1970, when Goldwater Memorial ceased to be a research institution, but it had a long-term impact on hospital-based education. When Goldwater Memorial closed in 2013 the range of educational programs transferred to successor institutions included “community-based English as a Second Language programs, General Educational Development training, higher education programs and vocational services” [35].

Gall’s Goldwater Memorial program is a comprehensive model for the education of bedbound individuals that provides a potential alternative to our current patchwork approach. First and foremost, the model emphasizes supported lifelong learning based on individual interest for personal growth as part of a total patient-care package, largely decoupled from the credential and course-based emphasis of public education. Gall also insisted on creating what we would currently call “a community of learners” as a critical element [34]. This represents at the foundation a new pedagogical approach to educating bedbound ME/CFS patients, especially when updated with new technological capacities; lessons from home-schooling for chronically ill children; advances in “gamification” of curriculum and new language-learning strategies; and innovative approaches to designing learning spaces for bedbound students.

Additionally, diverse elements outside American public education seem almost to have been crafted toward Gall’s model, or a close variant. There is a robust body of research on the “gamification of education,” with a significant subset focused on meeting the needs of special needs children. Gamification, although it has been variously defined since the first use of the term in 2008, generally refers to the practice of using elements of structure and rewards from digital gaming to support students in solving problems and remaining motivated, and experts constantly reiterate that “in gamification applications, students need not have to have toys, electronic devices, etc., and not always play games in order to learn” [36,37]. Materials presented via gamification are more likely to seamlessly meld short-term and long-term rewards for mastery, and are significantly more adaptable to the kinds of learning differences that chronically ill children (including those suffering from ME/CFS) manifest than traditional online courses [38,39,40]. Some examples include Kahoot, Quizizz, Quizlet and Gimkit. Transforming a classroom environment using gamification elements can enhance student learning and increase student understanding of the subject matter in a way that is enjoyable for students. For our bedbound and housebound patients, this brings creativity, play and collaboration into their lives [41].

A second example addresses the criticality of establishing a learning community to provide long-term psychosocial support for bedbound students. There is an understandable tendency in American society—especially after a year spent dealing with the COVID-19 pandemic—to consider virtual support communities as the default option here. In the Kohzikode district of the State of Kerala in India, however, education professionals are exploring a different strategy, that of physically constructing “resource rooms” in the houses of chronically bedbound students “to help differently abled bedridden students develop social and other skills” [23]. Besides computers, these rooms may contain “an FM radio, daily living aids, reading and writing materials, education and play materials, therapy balls, therapeutic seats, coloring books, and crayons” [23].

What makes these home-learning resource centers particularly innovative is that they are specifically linked to building social ties to a friendship peer group “from the school where the differently abled child is enrolled. They interact with the child during weekends or holidays to help them improve their skills. Resource teachers plan activities for the children keeping in mind their physical and mental condition” [23]. This emphasis on learning communities integrates well with new findings in regard to the impact of severe ME/CFS on children’s sense of self, and the emerging use of a “social,” or constructivist, approach to the teaching, assessment, and support of special needs students [13,42].

## 4. Moving forward to Enrich the Lives of Children Bedbound with Severe ME/CFS

In terms of education, bedbound children, adolescents, and young adults suffering from severe ME/CFS currently exist in a lonely frontier at the borders of public education, long-term care, psychotherapy, technological innovation, and communities too often completely unaware of any obligation to citizens they never see. The problem is not that we do not know enough about the requisite elements to enrich these young people’s lives through education, but that there is currently no framework to coordinate and deploy the knowledge, skills, supports, and technologies which already exist. Not only does the model of best practices have to be built, but we—as a society—have to figure out who will be responsible for enacting it.

There is reason to be hopeful, however. Slightly more than a decade ago we faced a similar state of affairs in supporting children with mild-to-moderate cases of ME/CFS to be educationally successful within the public schools. A broad coalition of researchers, clinicians, educators, parents, and volunteers coalesced to address that challenge, and they have made major progress. Students with ME/CFS increasingly find that schools are better prepared to teach them on their own terms. That process began with a few educators raising the issue, followed by a national dialogue among the concerned parties. Despite the loss of the Chronic Fatigue Syndrome Advisory Committee at the Department of Health and Human Services, which coordinated many of those early discussions, there are today a number of robust networks that cross disciplinary boundaries and are capable of initiating such conversations.

## Data Availability

Not applicable.

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
