# Peer review of "The Impact of Severe ME/CFS on Student Learning and K–12 Educational Limitations"

_healthcare, 2021, doi:10.3390/healthcare9060627_

Round 1

Reviewer 1 Report

This is an excellent brief review of the impact of ME/CFS on student learning in young people.   The author presents a cogent analysis of the problem and offers some specific and tangible examples for improvement.   The article fits perfectly within the theme of the special issue will be a wonderful addition to the scope of work likely to be offered.  I have only one minor edit, which is to make "flu-like symptom" in section 2 line 4.

Author Response

Thank you very much.  I corrected the grammatical error.  I appreciate the response with the reasons why liked the article.

Thank you for agreeing to do this. 

Faith Newton

Reviewer 2 Report

The article clearly communicates important concerns about student learning related to the education of housebound and/or bedbound students. The negative impact of extensive social and intellectual isolation for these students who are capable of learning is addressed while also outlining serious concerns related to educator training and the alarming absence of standards of care and best practices. Clear suggestions and examples are provided to demonstrate there is a reason for hope because we have the ability to resolve these concerns for students who need and deserve to benefit from targeted efforts to enrich their lives. Hopefully, shedding light on the concern via this article could be a start to evoking conversation followed by serious problem-solving efforts to address this dire need for homebound/bedbound students effectively.

Author Response

Thank you very much for the review.  I appreciate the positive comments. 

Faith

Reviewer 3 Report

This is a well written, informative article about the potential ways to address the needs of students with severe ME/CFS, written by an expert in the educational needs of such patients. I had only one minor suggestion for change, on page 2, in the second sentence of section 2. I would suggest an expansion of the definition of post-exertional malaise. Currently, this sentence states: “Post-exertional malaise (i.e. prolonged flu like symptom) and persistent fatigue make school attendance irregular …” I would suggest stating something like the following: Post-exertional malaise is a defining feature of ME/CFS, and is characterized by the provocation of fatigue as well as other symptoms (cognitive fogginess, headache, lightheadedness, flu-like symptoms, muscle aches and others) after increased physical, cognitive, or orthostatic stress. Poste exertional malaise and persistent fatigue make school attendance irregular …”

Author Response

Thank you it was a good suggestion. I add some additional language.  I appreciate you taking the time to review this article.  Have a great day!

Faith Newton
